

# Comparison of machine learning techniques for reservoir outflow forecasting

Orlando García-Feal[1,2], José González-Cao[1], Diego Fernández-Nóvoa[1], Gonzalo Astray Dopazo[3], Moncho Gómez-Gesteira[1]

[1]Centro de Investigación Mariña, Universidade de Vigo, Environmental Physics Laboratory (EPhysLab), Campus Auga, Ourense, 32004, España
[2]Water and Environmental Engineering Group, Department of Civil Engineering, Universidade da Coruña, A Coruña, 15071, España
[3]Universidade de Vigo, Departamento de Química Física, Facultade de Ciencias, 32004 Ourense, España

*Correspondence to*: Orlando García Feal (orlando@uvigo.es)

**Abstract.** Reservoirs play a key role in many human societies due to their capability to manage water resources. In addition to their role in water supply and hydropower production, their ability to retain water and control the flow makes them a valuable asset for flood mitigation. This is a key function since extreme events have increased in the last decades as a result of climate change, and therefore, the application of mechanisms capable of mitigating flood damage will be key in the coming decades. Having a good estimation of the outflow of a reservoir can be an advantage for water management or early warning systems. When historical data are available, data-driven models have been proven a useful tool for different hydrological applications. In this sense, this study analyses the efficiency of different machine learning techniques to predict reservoir outflow, namely multivariate linear regression (MLR) and three artificial neural networks: multilayer perceptron (MLP), nonlinear autoregressive exogenous (NARX) and long short-term memory (LSTM). These techniques were applied to forecast the outflow of eight water reservoirs of different characteristics located in the Miño River (northwest of Spain). In general, the results obtained showed that the proposed models provided a good estimation of the outflow of the reservoirs, improving the results obtained with classical approaches such as to consider reservoir outflow equal to that of the previous day. Among the different machine learning techniques analyzed, the NARX approach was the option that provided the best estimations on average.

## 1 Introduction

Humankind has been creating reservoirs since ancient times (Baba et al., 2018). The purpose of these bodies of water are varied and include irrigation, protection, power generation and control of the natural flow of rivers, among others (Lee et al., 2009). Reservoirs are created through the construction of dams, which are complex structures that retain water and are capable of controlling the water flow. This ability to control the flow permits the management of water resources allowing the storage of water for consumption, electricity production and protection against floods. They are, therefore, important agents that affect



the economy, human population and fauna and flora in their area of influence (Castelletti et al., 2008). In a river basin, it is usual to find several of these structures along the course of the river, thus, the operation of the upstream reservoirs affects all the downstream activities. Therefore, communication and coordination among the different dams present in the river course are desirable for the optimal management of water resources (Marques and Tilmant, 2013; Jeuland et al., 2014; Quinn et al.,

2019; Rougé et al., 2021). However, this is not always possible as there may be different barriers and tradeoffs that hinder such coordination. Is not unusual for rivers to pass through different countries or administrative regions with different policies and regulations. It is also common for dams to be operated by private companies with different operating policies and interests. The operation of these structures depends not only on natural factors and well-defined operating rules but also on external demand. This adds a significant amount of uncertainty in predicting the outflow of a reservoir at any given time, making it

difficult to incorporate into physics-based models, which is a disadvantage in water resource management and flood risk prevention.

Different aspects of water reservoirs will be of increasing importance in the future. One of the most important is the key role that dams play in protection against floods, which are one of the most dangerous natural catastrophes, being the cause of

tremendous loss of lives (Jonkman, 2005) and billions of euros in economic losses (Hallegatte, 2012; Wallemacq et al., 2018) worldwide. Unfortunately, several studies predict a worsening scenario for the future, increasing the frequency and severity of these phenomena (Berghuijs et al., 2017; Passerotti et al., 2020). Several factors affect this trend, being climate change (Arnell and Gosling, 2016; Liu et al., 2018) and modifications in land use (Booth and Bledsoe, 2009; Bradshaw et al., 2007; de la Paix et al., 2013; Rosburg et al., 2017) two of the most important ones. In response to these worrying reports, the scientific

community established that flood mitigation is one of the most important challenges to be addressed in the coming decades (Field et al., 2012), and dams can play a key role in this sense. In addition, in the context of climate change, hydropower generation also plays an important role. Hydropower generation is expected to increase significantly in the future in a scenario of increasing demand for renewable energies (Adaramola, 2016), albeit the climate change may affect river flow and thus the availability of water for power generation (Berga, 2016). Issues related to water availability in future scenarios are also a major

concern for such important sectors as agriculture, which negatively affect food production (Alcamo et al., 2007; Elliott et al., 2014; Xiong et al., 2009). In this context, to mitigate these effects it will be essential to optimize the management of water resources, and more specifically the water reservoir operations.

This research paper focuses on the development of models capable to forecast the outflow of a reservoir. These models could

be an advantage when incorporated into current or newly developed water management systems, improving their operation. This includes flood early warning systems and reservoir management systems, among others. In order to forecast the outflow of a reservoir, the most simplistic approach involves assuming that the reservoir is at 100% of its storage capacity. Therefore, the lamination capacity of the dam is almost cancelled, and the outflow is equal to the inflow. Although this approach is an





over-simplification of river dynamics, it can be a good approximation in flood scenarios during wet seasons, especially in
small reservoirs or when they are nearly full and have little margin to alter the natural flow of the river. Under normal
conditions, the simplest approximation is to assume that the outflow of the reservoir for a given day *d* will be the same as on
day *d-1*. This can provide acceptable approximations under normal conditions when the flow does not vary significantly from
day to day. This procedure can be improved by applying multivariate solutions that assume different weights to several known
variables. In this case, the approach will be to establish a relation between the outflow on a given day *d* with the known outflow,
inflow and filling dam level on day *d-1*. One approach to improve these simpler solutions is to develop data-driven models
based on the analysis of the data of a specific system, being able to find relations between input and the output variables of the
system. These models have been complementing or replacing physics-based models (e.g., hydrodynamic models) in the last
years (Solomatine and Ostfeld, 2008). Data-driven modelling uses machine learning (ML) techniques to build models for a
specific system from existing data. According to Rashidi et al. (2019), machine learning is an application of artificial
intelligence (AI) that enable the automatic learn of computer systems, all based on experience without explicit programming.
Machine learning techniques were successfully applied in many hydrological applications (Le et al., 2019; Xiang et al., 2020;
Kratzert et al., 2018; Rjeily et al., 2017; Guzman et al., 2017; Lee and Tuan Resdi, 2016; Taghi Sattari et al., 2012; Ghorbani
et al., 2019; Emami and Parsa, 2020; Sammen et al., 2017) in the last years. In this paper, several methodologies based on
machine learning techniques are proposed for the time series forecasting of reservoir outflow. The ML techniques used for this
task are included under the category of supervised learning, meaning that the ML algorithm will be fed with data that includes
the desired solutions (Géron, 2019), in this case, the future outflow of a reservoir. The task to be performed by the ML model
is a regression in which a target outflow value will be predicted from a series of input variables or features. This research paper
will cover several techniques ranging from multivariate linear regression (MLR) to several artificial neuron network (ANN)
techniques (a feed-forward neural network with back propagation algorithm (multi-layer perceptron, MLP), nonlinear
autoregressive exogenous (NARX) and a long-short term memory network model (LSTM)) to forecast the outflow for different
important dams located in the Miño-Sil basin (Galicia, Spain).

The research paper organizes as follows: in section 2 the characteristics of the area of study will be presented as well as the
different ML models employed. Section 3 will analyse and discuss the results obtained. And last, in section 4 the main
conclusions of the study will be exposed.

## 2 Material and methods

### 2.1 Area of study

Figure 1 shows the area of study and the location of the reservoirs employed in the present work. The eight reservoirs are
located in the Miño-Sil River basin, situated in the northwestern of the Iberian Peninsula. The basin has a total area of around





17,000 km$^2$ (Confederación Hidrográfica del Miño-Sil, 2016) and constitutes an important region of hydroelectric generation. The Miño-Sil river system is one of the most important in the Iberian Peninsula and the one with the highest runoff-to-surface ratio. It is characterized by a pluvial regime with the maximum water flow in the winter season and a minimum in summer (Fernández-Nóvoa et al., 2017), presenting average annual precipitation of 1,184 mm (Confederación Hidrográfica del Miño-Sil, 2016). Eight reservoirs were selected from the Miño-Sil river system with capacities ranging from 10 to 655 hm$^3$ (see

Table 1 for a summary of their main characteristics).

A total of 19 years of data in a daily scale were provided, after request, by the Minho-Sil River Basin Authority (Confederación Hidrográfica del Miño-Sil, https://www.chminosil.es) for the reservoirs under study. The period analyzed spans from October 1, 2000, to September 30, 2019. The time series data include the percentage of filled volume, the inflow and the outflow of the

reservoir.

## 2.2 Machine learning models

The available dataset was divided into three different subsets, using roughly the first 70% of the data range for the training subset (from October 1, 2000, to September 30, 2013), the following 15% for the validation subset (from October 1, 2013, to September 30, 2016) and the remaining 15% for the test subset (from October 1, 2016, to September 30, 2019). This criterion

has been chosen for better interpretation and comparison of the output series produced by the models. Table 2 shows the statistics of the subsets for each reservoir and variable. Once the training phase is completed, the unbiased model performance is tested against the test subset. This approach can help to identify overfitting problems, where a model offers good performance with the dataset used during the training phase but is not able to generalize with new input data. For all the models, the inflow, outflow and volume percent values are used as input data to predict the outflow of the next day.

**2.2.1 Multivariate linear regression**

The first ML technique based on multivariate linear regression was chosen to test complex neural network-based techniques against more conventional techniques. MLR can perform better than ANN in certain applications where the sample data available is small (Markham and Rakes, 1998), in this sense, this model can help to assess if the dataset available is big enough to use ANN based models. A relation was established between the outflow and inflow measured at day $d$, respect to the outflow

of the next day $d+1$, which corresponds to the day under prediction. In addition, this adjustment was carried out not only for each dam but also for different filling levels, in order to take also into account this variable, which plays a key role in the dam capacity to retain water. In this case, percentage sections of 10% filling of the dam were considered, which means an adjustment when the occupied volume of the dam is less than 10%, another when the occupied volume is between 10-20%, and so on.



Thus, for each dam, 10 adjustments define the outflow prediction attending to the different level of occupation, as indicates
the next equation:

$$\hat{y}_{d+1} = c_0 + c_1 r_d + c_2 y_d \tag{1}$$

where $\hat{y}_{d+1}$ is the predicted outflow for day $d+1$, being $y_d$ and $r_d$ the measured outflow and inflow for day $d$. The three
coefficients ($c_0$, $c_1$ and $c_2$) were obtained from the linear fitting depending on the measured filling level of the corresponding
dam.

The equation and procedure described above were applied to each dam, using in all cases the first 70 % of the data to obtain
the adjustments, and the last 30 % to test their efficiency. Although a validation phase is not considered in this methodology,
the same dataset partitioning as in the neural network models has been used to facilitate the comparison. The first 70 % of the
data (training subset) will be used to develop the model, and the last 30 % to test (validation and test subsets) their efficiency.
Therefore, both the validation and test subsets are both test subsets in this methodology.

### 2.2.2 MLP Model

The second type of model developed in this research was ANNs, which are a type of computational approach that can be
encompassed within machine learning. These types of models have been inspired in the biological human brain (Farizawani
et al., 2020). An ANN model, like a biological neural network, is formed by several simple processing units joined to each
other using weighted connections (Taghi Sattari et al., 2012). The simple processing unit is called node or neuron. Artificial
neural networks present different advantages over traditional approaches to model data (Livingstone et al., 1997). Perhaps, and
according to Livingstone et al. (1997), the most outstanding advantage is that this type of model is capable to fit complex
nonlinear models. However, according to the same authors, this type of approach also has an important disadvantage, which
is that neural networks can suffer from overfitting and overtraining but can be solved by taking a good architecture selection
and using training/control groups to see the evolution of the model (Livingstone et al., 1997).

The first type of ANN model developed in this research is an MLP-ANN (multi-layer perceptron, artificial neural network),
that is, a feed-forward neural network with a back propagation algorithm. In this type of ANNs, the information moves only
in forward direction, that is, from the input neurons (in the input layer), crossing the hidden neurons (in the hidden layer) to
the output neurons (RapidMiner Inc., 2022) (see Figure 2). The back propagation algorithm is used to fit the model. This kind
of supervised algorithm compares the predicted values with the real values to calculate the prediction error, then this error is
fed back through the network to adjust each connection weight and reduce the prediction error in the next cycle (RapidMiner
Inc., 2022), that is, this ANNs learn by adjusting the connection weights. This process continues to run until the error goes



down or reaches a satisfactory level, or until a previously established number of cycles has been reached (Taghi Sattari et al., 2012).

This type of artificial neural network is widely used in different predictions related to the study of water movement and dam
or reservoirs management. In this sense, an MLP model, with back propagation learning algorithm, has been used to model the daily inflow into the Eleviyan reservoir (Iran) (Taghi Sattari et al., 2012). This model has been compared with a time lag recurrent neural network (TLRN) for a period between 1 September 2004 and 30 June 2007. The MLP models were developed with only one hidden layer. According to Taghi Sattari et al. (2012), both models work acceptably with low inflow values, besides, the performance of both models was very close. Another interesting study where MLP was used is in the research
carried out by Ghorbani et al. (2019) who designed and evaluated a hybrid forecasting model combining a gravitational search algorithm (GSA) with an MLP to predict the water level in two lakes (Winnipesaukee and Cypress, USA). This hybrid model is compared with an MLP model that used the Levenberg-Marquadt back propagation learning algorithm and other models; hybrid models (MLP-Particle Swarm Optimization and MLP-FireFly Algorithm) and ARMA and ARIMA models (Ghorbani et al., 2019). The hybrid MLP-GSA model showed a high efficacy over the other developed models and suggests that can be
used in water resources management among other tasks. On the other hand, reservoir storage capacity determination is an important element in water resources management and planning, among others (Emami and Parsa, 2020). Due to this, Emami and Parsa (2020) try to predict the optimal reservoir storage capacity, using an evolutionary algorithm (inspired by imperialistic competition) along with an MLP model with a back propagation training technique, applied to Shaharchay dam (Urmia Lake basin, Iran). According to the results, both models are satisfactory (with RMSE of 0.041 and 0.045 for the imperialist
competitive algorithm and the ANN model, respectively) (Emami and Parsa, 2020). Finally, another interesting research is the one carried out by Sammen et al. (2017) which used a generalized regression neural network (GRNN) to predict the peak outflow in the event of a possible dam failure. Sammen et al. (2017) built six models using different dam and reservoir attributes and concluded that the GRNN model show potential to predict peak outflow.

As previously said, the first ANN models developed were feed-forward neural networks with a back propagation algorithm. In this kind of ANNs, the information passes through different layers. In the input layer, the information is received from the database, and it is sent to the hidden layer where the information is treated. Finally, this new information is sent to the output layer where a result is generated. The number of neurons in each layer is determined by the nature of the problem. In this sense, the number of neurons in the input layer is fixated by the number of variables (inflow, outflow and volume (%) at day
$d$) that will be used to try to predict the desired variable (outflow for day $d+1$). In the output layer, there will be as many neurons as variables to be predicted (in this case, one). Finally, in this research, only one hidden layer was used, and the number of neurons was determined by the trial-and-error method (being studied between one and seven). The number of cycles was studied between 1 and 131,072 in 17 steps with a logarithmic or lineal scale, and the decay parameter was used to decrease the


learning rate during the learning process (true or false). The best MLP model developed (lineal or logarithmic scale) was
selected based on the lowest RMSE value for the validation subset.

The different MLP models were implemented in a server (AMD Ryzen 7 1800X, Eight-Core Processor 3.60 GHz with 16GB
of RAM) located at the Department of Physical Chemistry of the University of Vigo, Campus of Ourense. The operative
system used was Windows 10 Pro 20H2 with 64-bit. The MLP models were developed using RapidMiner Studio 9.8.001
software.

### 2.2.3 NARX model

The last two ANN techniques analysed fall under the umbrella of the so-called recurrent neural networks (RNN). This kind of
ANN is especially suitable to forecast time series. Nonlinear autoregressive with exogenous inputs (NARX) neural networks
are a type of RNN designed for tasks with long-term dependencies on the input data. They can converge and generalize faster
than other ANNs (Lin et al., 1996). NARX can use previous input and output data including a feedback delay for both input
and output. There are two typical NARX model architectures: parallel (P) and series-parallel (SP) (Xie et al., 2009) see Figure
3. In the first one, the output of the model is fed back into the neural network, whereas in the SP architecture the real output
value is used during the training phase. This second approach has proven to be more stable and robust (Amirkhani et al., 2022;
Narendra and Parthasarathy, 1990). NARX models were used in multiple hydrological applications, a NARX model was used
to forecast flood risks in urban drainage systems in Rjeilhay et al. (2017), showing a good performance and better calculation
speed compared with physically based models. Guzman et al. (2017) developed a NARX model to simulate groundwater levels
in the Mississippi River Valley Alluvial aquifer (USA). Lee and Tuan Resdi (2016) developed a NARX model capable to
make hydrological prediction at multiple gauging stations in the Kemaman catchment (Malaysia) using thirteen meteorological
input parameters. Yang et al. (2019) compared several RNN models including LSTM, NARX and a genetic algorithm based
NARX for reservoir operation using input data from a distributed hydrological model.

A general NARX model following the SP architecture can be expressed as

$$\hat{y}_{d+1} = f\left(y_d, y_{d-1}, \dots y_{d-n_y}, x_d, x_{d-1}, \dots, x_{d-n_x}\right) \tag{2}$$


Where $\hat{y}_{d+1}$ is the predicted outflow for day $d+1$, $y_d$ is the measured outflow for the day $d$, $x_d$ are the exogenous input
variables for day $d$ and $f$ is a nonlinear function that is approximated by a MLP. The parameters $n_x$ and $n_y$ refer to input and
output delays. In this work, the values of $n_x$ and $n_y$ were obtained using cross-correlation functions and the value for both
parameters was set equal to 5 days. The value of the number of hidden neurons was set equal to 8 by the trial-and-error method.





The activations functions for the hidden layer of the neural network and the output layer are tan-sigmoid and linear, respectively. The Levenberg-Marquardt algorithm was defined to train the model using the mean square error (MSE) as the loss function. The NARX models were developed using MATLAB software (MathWorks Inc., 2022).

### 2.2.4 LSTM model

The Long Short-Term Memory (LSTM) first proposed by Hochreiter and Schmidhuber (1997), employs the so-called LSTM
cell that it is a type of RNN memory cell that stores a short-term state $h_d$ and a long-term state $c_d$. It is capable of identifying meaningful input data and storing it in a long-term state, keeping this data as long as necessary and using it when needed. Due to this fact, this approach is very suitable to capture long-term patterns present in time series. As shown in Figure 4, for each time-step, as the $c_{d-1}$ state enters the cell, some data is dropped in the forget gate, to add some extra data coming from the input gate, resulting in $c_d$. At the same time, a *tanh* function is applied to $c_d$ and passes through the output gate to produce $h_d$ that is
equivalent to $\hat{y}_{d+1}$. The input data $x_d$ and the short-term state $h_{d-1}$ are fully connected to four layers. The main one, which outputs $g_d$, analyses the input and the previous short-term state as in a regular RNN but only the most important parts are stored into the long-term state (Géron, 2019):

$$g_d = tanh\big(W_g[h_{d-1}, x_d] + b_g\big) \tag{3}$$


where *tanh* is the activation function, $W_g$ is a weight matrix and $b_g$ is a bias matrix corresponding to the main layer. The remaining three layers are the gate controllers that use a sigmoidal activation function. Their outputs range from 0 to 1 and, since these outputs are used in an element-wise product, they have the ability to open or close the gate. The forget gate that outputs $f_d$ controls which part of the long-term state will be erased:


$$f_d = \sigma\big(W_f[h_{d-1}, x_d] + b_f\big) \tag{4}$$

where $\sigma$ is the sigmoid activation function. The input gate, that outputs $i_d$, controls which parts of the main layer will be added to the long-term state:


$$i_d = \sigma(W_i[h_{d-1}, x_d] + b_i) \tag{5}$$

The last gate is the output gate, that outputs $o_d$ and controls which parts of the long-term state should be included in this time step $h_d$ and $\hat{y}_{d+1}$:




$$o_d = \sigma(W_o[h_{d-1}, x_d] + b_o) \tag{6}$$

Therefore, the cell output and the new short- and long-term states are defined as follows:

$$c_d = f_d c_{d-1} + i_d g_d \tag{7}$$
$$\hat{y}_{d+1} = h_d = o_d tanh(c_d) \tag{8}$$

LSTM models were used in multiple hydrological applications. Le et al. (2019) used a LSTM neural network for flood forecasting in the Da River (Vietnam) using precipitation and flowrate at daily scale as input data, achieving predictions for one, two and three days ahead showing very good NSE values of 99%, 95% and 87% respectively. Xiang et al. (2020) used a LSTM and seq2seq (Sutskever et al., 2014) modelling approach to estimate 24h rainfall-runoff on an hourly scale in Clear Creek and Upper Wapsipinicon River watersheds (Iowa, USA), using observed and forecasted rainfall, observed runoff and evapotranspiration data obtained from stations. Results showed that the methodology could provide effective predictions for hydrology applications. Kratzert et al. (2018) used data from 241 catchments from the CAMELS data set to compare LSTM as a hydrological model versus a more traditional approach using the Sacramento Soil Moisture Accounting Model (SAC-SMA) coupled with Snow-17, obtaining similar results with both approaches. Zhang et al. (2018) use a LSTM model to simulate the reservoir operation using 30 years of data from the Gezhouba dam located on the Yangtze River (China), outperforming other machine-learning approaches such as a back-propagating neural network and a support vector regression.

The software used for the implementation of the LSTM models was TensorFlow (TensorFlow Developers, 2022). After an exploration of different hyper-parameters, the number of hidden layers was set to one as no significant benefit was observed when using a higher number. The input width window chosen was 10 days. The input data was scaled using a standard scaler. The optimizer chosen was Adam with Nesterov momentum (Kingma and Ba, 2014; Dozat, 2016). The batch size was set to 32, obtaining similar results with values between 16 and 32 and achieving significantly worse results with higher values, these observations are in concordance with existing literature (Masters and Luschi, 2018). The activation function used is *tanh*, since it was the only option that was optimized for GPU (Graphics Processing Unit) accelerators giving much faster training times but also gives the more consistent results. The number of training iterations was defined by an early stopper function that stops the process when no further improvements are obtained. The cost function used during the training was the mean square error (MSE). On the other hand, the number of neurons and the learning rate parameters were optimized for each specific reservoir using the KerasTuner software (O'Malley et al., 2019).





**2.3 Metrics**

In order to compare the different models and measure their accuracy several statistical metrics, widely used in hydrological applications, were employed. More precisely, the Pearson's coefficient of correlation ($r$), the ratio of root mean square error and the standard deviation of the observed values (RSR), the Nash-Sutcliffe Efficiency coefficient (NSE) (Nash and Sutcliffe,

1970) and the percent bias (PBIAS) that are defined by the following equations:

$$r = \frac{\sum_{i=1}^{N}\left[\left(Q_i^{for}-\overline{Q^{for}}\right)\left(Q_i^{obs}-\overline{Q^{obs}}\right)\right]}{\sqrt{\sum_{i=1}^{N}(Q_i^{for}-\overline{Q^{for}})^2}\sqrt{\sum_{i=1}^{N}(Q_i^{obs}-\overline{Q^{obs}})^2}} \tag{8}$$

$$NSE = 1 - \frac{\sum_{i=1}^{N}(Q_i^{obs}-Q_i^{for})^2}{\sum_{i=1}^{N}(Q_i^{obs}-\overline{Q^{obs}})^2} \tag{9}$$

$$RSR = \frac{\sqrt{\sum_{i=1}^{N}(Q_i^{obs}-Q_i^{for})^2}}{\sqrt{\sum_{i=1}^{N}(Q_i^{obs}-\overline{Q^{obs}})^2}} \tag{10}$$

$$PBIAS = \frac{\sum_{i=1}^{N}(Q_i^{obs}-Q_i^{for})}{\sum_{i=1}^{N}(Q_i^{obs})} \times 100 \tag{11}$$

where $Q^{for}$ is the forecasted value, $Q^{obs}$ is the observed value and N is the total number of samples. Following the criterion of Moriasi et al. (2007), the statistics for model evaluation can be divided into three categories. First, the standard regression statistics that measure the linear relationship between the predictions made by a model and the observed data, in this category

we considered the Pearson's coefficient of correlation ($r$), which ranges between -1 and 1, with values close to 1 being considered as a high degree of positive linear relationship. The second category is the dimensionless statistics, for this case we have chosen the NSE, which ranges from -∞ and 1.0, it is a normalized statistic that computes the relative magnitude of the residual variance with respect to the variance of the observed data, with 1.0 being the optimal value. The last category is error index statistics that quantify the deviation of the predicted values compared with the observed values in the data units used. In

this last category, two statistics were chosen; on the one hand, the RSR is the ratio of the RMSE (root mean square error) to the standard deviation of the observed data, with 0 being the optimal value. The other error index statistic used is the PBIAS, which calculates the average tendency of the predicted values to underestimate (positive PBIAS) or overestimate (negative PBIAS) the observed series.

**3 Results and discussion**

Statistical parameters used to evaluate the performance of the models to predict dam outflow are shown in Table 3. MLP models with linear scale are showed in this table due to present slightly better average adjustments in the validation phase than the models with logarithmic scale. Results corroborate that the proposed models offer a "very good" performance for all the





subsets considered according to the criteria defined by Moriasi et al. (2007), who established different ranges of these statistical parameters to define the level of functioning of these procedures (the only exception is the PBIAS value obtained by the linear regression model in the training subset of the San Martinho reservoir). All the statistical parameters under consideration reach the maximum level of good functioning defined, and therefore it can be concluded that all the proposed models are able to provide an accurate prediction of dam outflow attending to known parameters.

The first approximation to the prediction of dam outflow was made by the simple method of considering as the prediction outflow the same outflow measured in the previous day. This is considered as the baseline model, on which the rest of the models will be compared. Figure 5 shows the reservoirs averaged metrics for each model and subset. As we can see in RSR, NSE and *r*, all the models improve the accuracy of the baseline model in every dataset. The ML model's performance shows no evidence of overfitting problems, where a trained model learns very specific features of the training dataset and fails to generalize using new datasets. The MLR approach was able to outperform the baseline model on the whole dataset but lags behind the ANN based models. All the ANN models showed similar accuracy based on RSR, NSE and *r* metrics and provide a good generalization across the test subset. The LSTM models were slightly better than MLP whilst NARX has shown the best performance.

As can be observed in PBIAS (Figure 5), the baseline does not offer any significant bias since it is the same data series as the one observed but with a delay. All the ML models have a tendency to overestimate the series, especially in the test subset. The LSTM models have the lowest tendency to overestimate and the MLR models have the highest. This fact can be very significant depending on the application of the model where a more conservative estimation towards the worst-case would be preferable.

In order to analyse the differences in the performance on the different datasets and reservoirs of each model, Figure 6 shows the NSE values for each case. Taking into account that the NSE metric is very popular among hydrology studies, and no significant differences were found with RSR and *r*, only the NSE is shown for sake of clarity. The MLR models provide a good generalization in Castrelo, Velle, Santo Estevo, San Martinho and Frieira reservoirs where the performance for the test dataset is very similar or even better than on the training dataset, these are low-capacity reservoirs (excepting the Santo Estevo reservoir) where the regulation capacity is also lower. This tendency is also present in the ANN models, where in the lower capacity reservoirs (Castrelo, Velle, San Martinho and Frieira) show a better generalization ability, whilst in the higher capacity reservoirs (Belesar, Barcena, Santo Estevo and Peares) the performance of the models in the test subset is lower than on train and validation subsets. Looking more closely at the data in Table 3 a tendency is detected in reservoirs with a higher capacity to have worse statistics than those of lower capacity. This evidences that higher capacity reservoirs have a greater ability to regulate the flow of the river according to the desired interest. In similar events, higher capacity reservoirs have more





possibilities of actuation, meanwhile lower capacity ones have a much more limited range of options. This particularity is more difficult to be modelled by the ML models.

The NSE values for the test dataset for each reservoir and model are shown in Figure 7, aiming to spot differences in the models performance depending on the reservoir. Belesar, Castrelo and Santo Estevo reservoirs show the highest advantage for

the ML models. On the opposite, in the Barcena reservoir, only NARX and LSTM were able to improve the baseline approach. The MLR models were able to outperform the baseline except in the Barcena and San Martinho cases, also in the Castrelo case improved the MLP model. The MLR was never the best option but usually provides results close to ANN models. The MLP models performed always better than the baseline model except in the Barcena case, they provide similar results to the other ANN models and very close to LSTM. The NARX models were able to improve the baseline model, they offered the best

results in all the reservoirs except the Barcena reservoir, standing as the best performer. The LSTM models also were able to consistently outperform the baseline model, being the best model in the Barcena reservoir and the second best model in the rest of the reservoirs, except in Santo Estevo where it was outperformed by the MLP and NARX models. From these observations, it can be concluded that a per-reservoir analysis is advisable when developing a data-driven model, since none of the methodologies proposed can be chosen as the best for all the cases.


To illustrate the behaviour of the developed models, the Belesar reservoir was chosen for two main reasons: it has the higher capacity from the analysed reservoirs, so it has a higher regulation capacity, and it doesn't have any other reservoir upstream that condition its behaviour. A comparison of the predicted and observed flow time series for the test period in the Belesar reservoir is shown in Figure 8. It can be seen that all the methodologies show similar performance, but some differences can

be highlighted. Both MLR and MLP models have underestimated the main peak of the series whilst NARX and LSTM models have overestimated it. This fact should be considered when designing systems like flood EWS where the worst-case estimation should be accounted for safety reasons. On the opposite, in the lower flows on the dry season, the NARX model was the best performer providing accurate predictions, however, the LSTM model had some difficulties at very low flow rates, especially in the summers of 2017 and 2019. This makes the LSTM model less suitable for water management systems where the accuracy

on the dry season is essential for a better exploitation of the water resources.

## 4 Conclusions

This research paper presents an assessment of different ML techniques applied to reservoir outflow one day ahead prediction using the previous reservoir volume percent, inflow, and outflow data. For this purpose, different models were developed and applied to several reservoirs in the Miño-Sil catchment. The analysis of the obtained results revealed that the proposed ML

techniques obtained accurate predictions. The ML models provide significant improvements over the baseline model showing a good generalization without significant signs of overfitting. On average, the MLR models were able to consistently improve



the baseline model, while the ANN models provided the best results. The RNNs (NARX and LSTM) improved the MLP results, showing the advantages of the RNNs when working with time series, especially in the case of NARX. When analysing the individual results for each reservoir, the NARX models obtained the best statistics except in the case of the Barcena

reservoir which was outperformed by the LSTM model, this evidences the need to perform a per-reservoir analysis to check the validity of each solution given the particularities of each reservoir.

Given that the ANN-based models were able to outperform the more traditional MLR models, it can be concluded that the number of samples in the dataset are within the limit for training an ANN, albeit larger datasets would possibly lead to better models especially when dealing with extreme events.

The overall observations confirm that the analysed ML models are capable to predict the outflow of reservoirs and, therefore, can be incorporated into different systems such as water resource management systems or early warning systems. Although the validity of these models was assessed using only a limited range of input variables, there are many other variables that influence the functioning of the reservoirs, such as the weather forecast or the electric power demand. It is being studied how to incorporate these variables into these models and if they can add any improvement to the predictions.


*Code and data availability:* The data are available upon request to the Confederación Hidrográfica del Miño-Sil.

*Author contribution:* OGF: Conceptualization, Methodology, Software, Investigation, Formal Analysis, Visualization and Writing – Original Draft Preparation. JGC: Conceptualization, Methodology, Software, Investigation, Formal Analysis and

Writing – Review & Editing. DFN: Conceptualization, Methodology, Software, Investigation, Formal Analysis and Writing – Review & Editing. GAD: Conceptualization, Methodology, Software, Investigation, Formal Analysis and Writing – Review & Editing. MGG: Conceptualization, Formal Analysis, Supervision and Writing – Review & Editing.

*Competing interests:* The authors declare that they have no conflict interest.

**ACKNOWLEDGEMENTS**

We especially thank to the hydrological planning office "Confederación hidrográfica Miño-Sil" for providing access to historical data without which it would be impossible to conduct this study.

The aerial pictures used in this work are courtesy of the Spanish Instituto Geográfico Nacional (IGN) and part of the Plan Nacional de Ortofotografía Aérea (PNOA) program.

This work was partially funded by the European Regional Development Fund under the INTERREG-POCTEP project RISC_ML (Code: 0034_RISC_ML_6_E). This work was also partially financed by Xunta de Galicia, Consellería de Cultura, Educación e Universidade, under Project ED431C 2021/44 "Programa de Consolidación e Estructuración de Unidades de Investigación Competitivas".



OGF was funded by Spanish "Ministerio de Universidades" and European Union – NextGenerationEU through the "Margarita Salas" post-doctoral grant.

DFN was supported by Xunta de Galicia through a post-doctoral grant (ED481B-2021-108).

GAD thanks to the University of Vigo for his last contract supported by "Programa de retención de talento investigador da Universidade de Vigo para o 2018" budget application 0000 131H TAL 641. GAD also thanks to RapidMiner Inc. for the different licenses of RapidMiner Studio software.

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



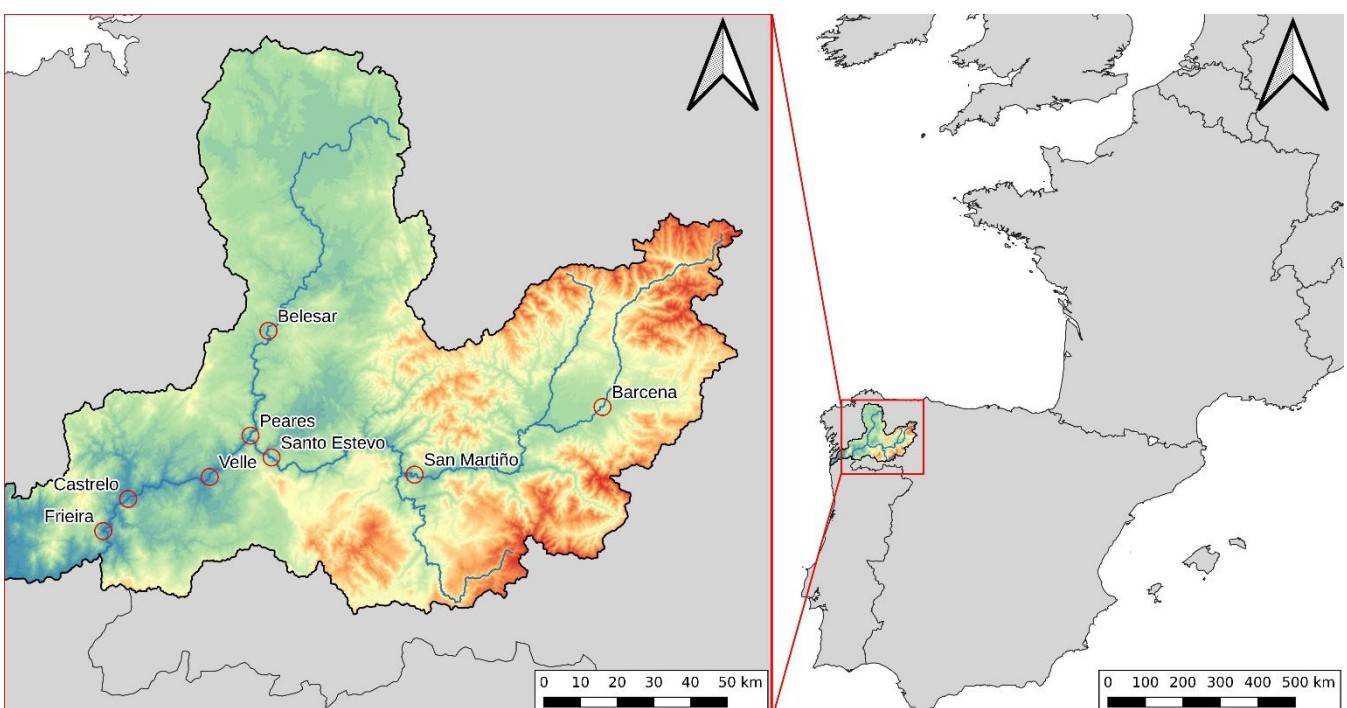

**Figure 1. Location of the reservoirs analysed in this study.**


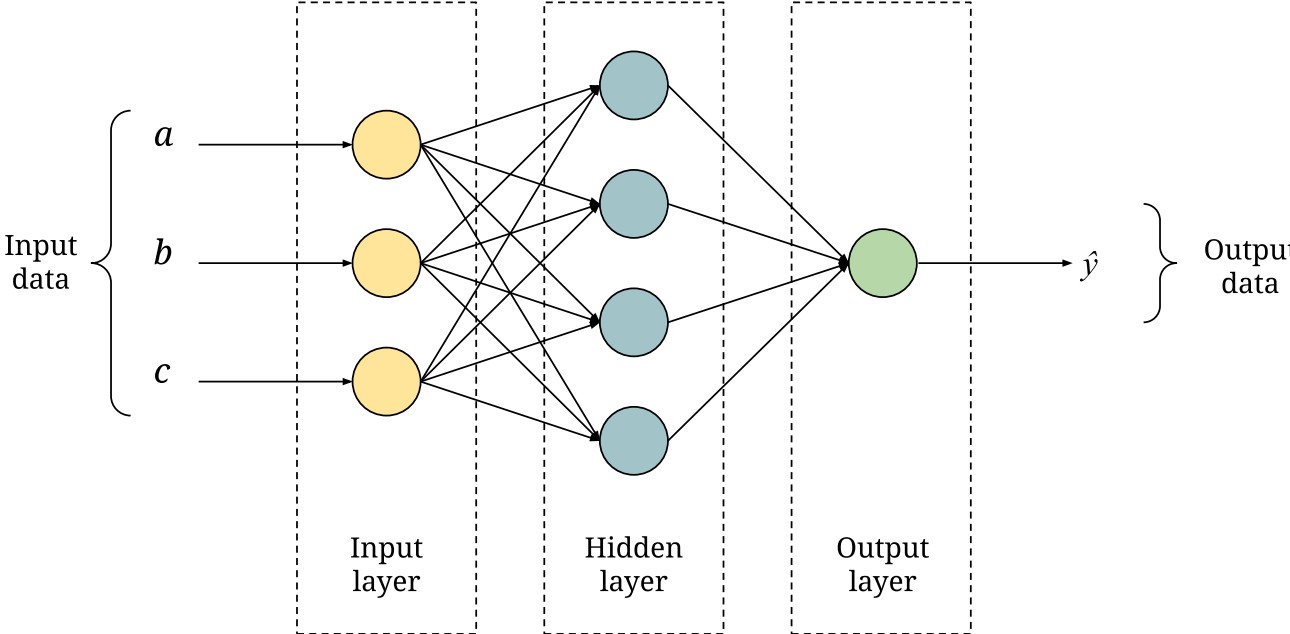

**Figure 2. Architecture of a multilayer perceptron with three inputs, a hidden layer with four neurons and a single output.**


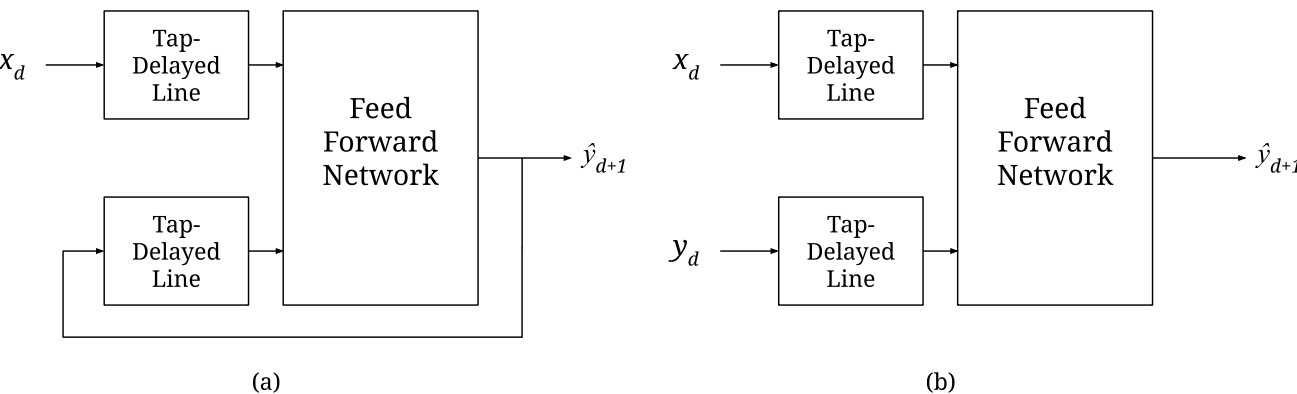

(a)

(b)

**565  Figure 3. Different architectures of a NARX model: (a) Parallel Architecture (b) Series-Parallel Architecture.**

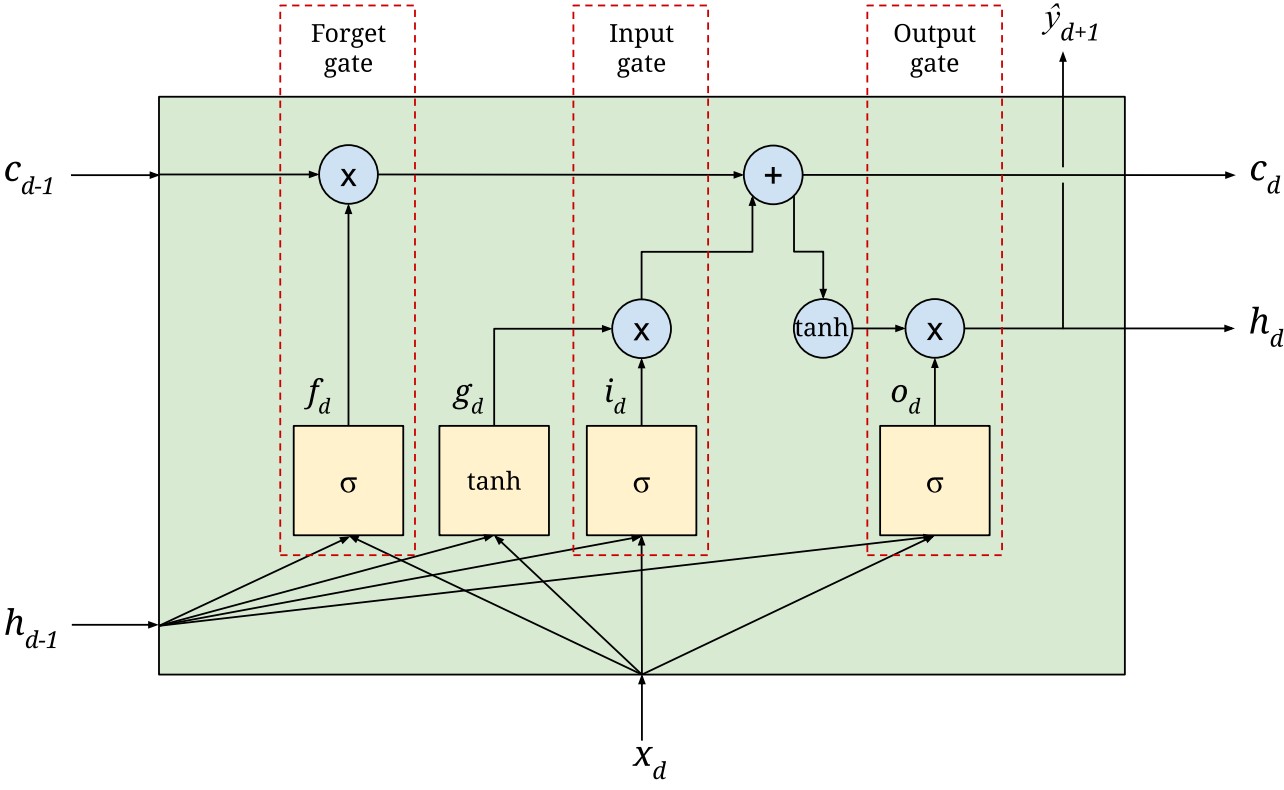

**Figure 4. Schematic view of a LSTM cell.**


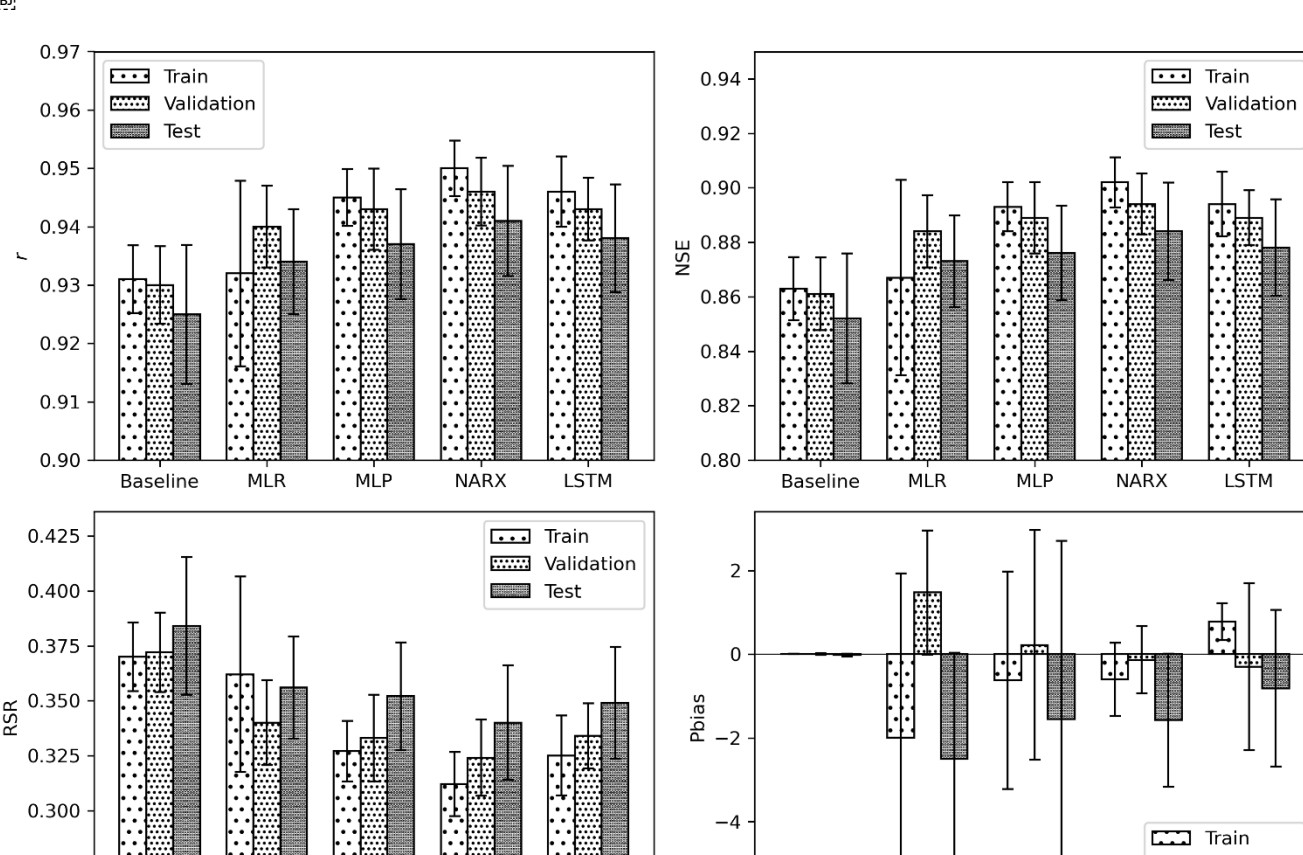


**Figure 5. Statistics of each subset for the different models developed. The average values from all the reservoirs are shown.**

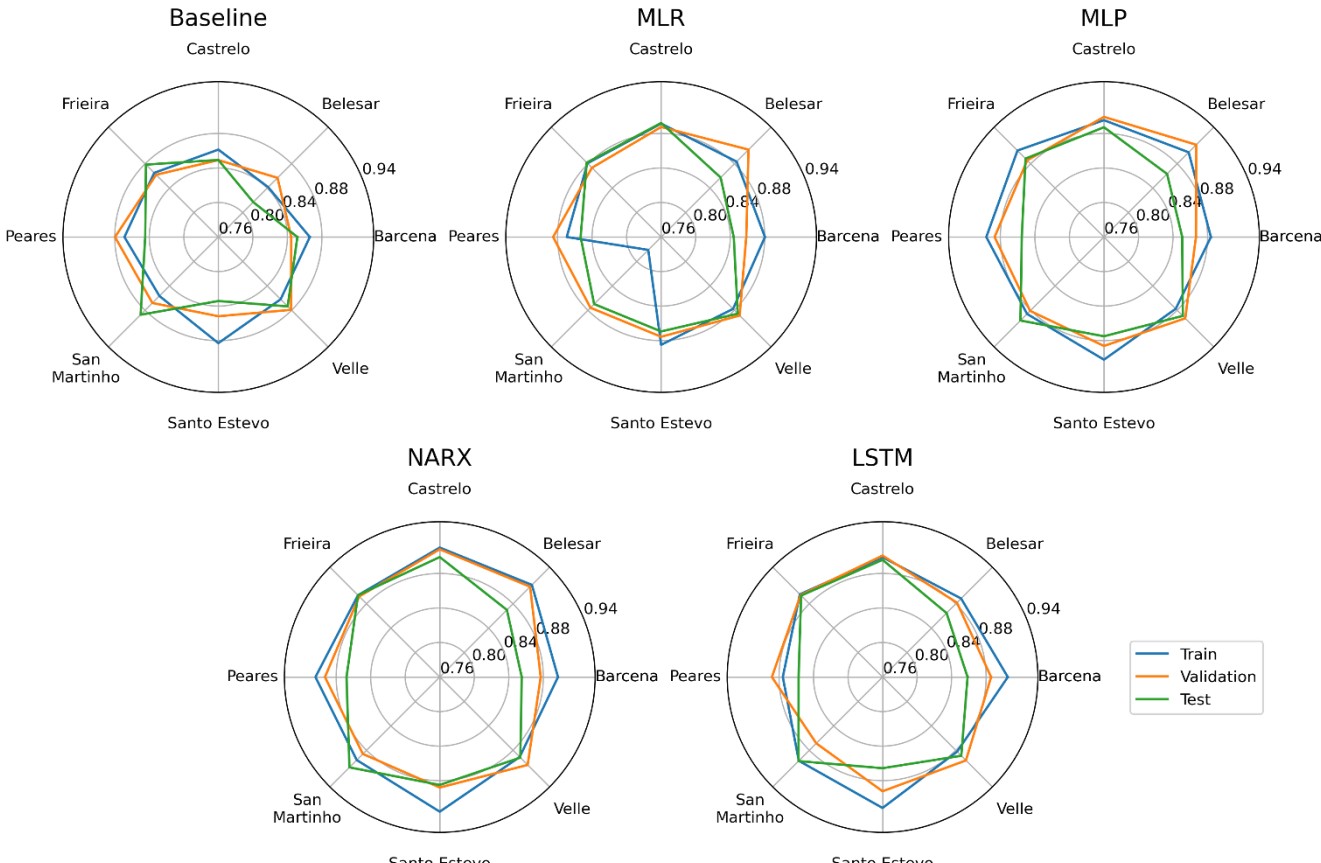

**Figure 6. NSE values for each dam obtained with five methodologies for train (blue line), validation (orange line) and test (green line).**


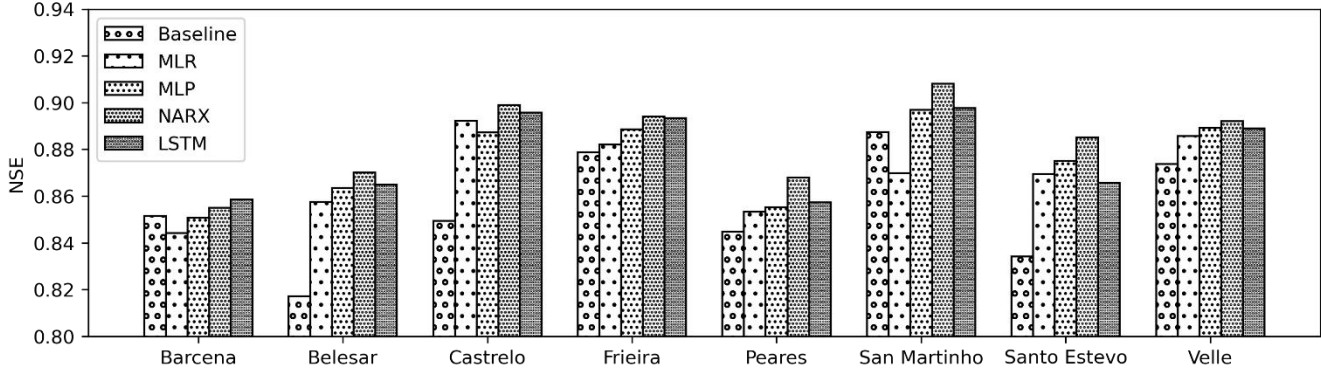

**Figure 7. Bar chart comparing the NSE values for the test dataset for each reservoir and model.**





**Figure 8. Time series (left) and scatter plots (right) for Belesar dam using the test dataset obtained with the proposed models.**

**Table 1. List of dams analysed in the study (data provided by Confederación Hidrográfica del Miño-Sil).**

| Reservoir Name | Capacity (hm$^3$) | Built on | Catchment area (km$^2$) |
|---|---|---|---|
| Belesar | 655 | 1963 | 4290 |
| Peares | 182 | 1955 | 4533 |
| Velle | 17 | 1966 | 12530 |
| Castrelo | 60 | 1969 | 13180 |
| Frieira | 44 | 1970 | 15160 |


| | | | |
|---|---|---|---|
| **Santo Estevo** | 213 | 1955 | 7216 |
| **San Martinho** | 10 | 1956 | 4740 |
| **Bárcena** | 341 | 1960 | 832 |


**Table 2. Statistics of the variables used for the train, validation and test subsets.**

| Reservoir | Variable | Train (2000-2013) | | | | Validation (2013-2016) | | | | Test (2016-2019) | | | |
|---|---|---|---|---|---|---|---|---|---|---|---|---|---|
| | | Mean | SD | Min. | Max. | Mean | SD | Min. | Max. | Mean | SD | Min. | Max. |
| **Barcena** | **Inflow (m³)** | 24.1 | 28.0 | -17.0 | 314.0 | 33.2 | 30.6 | -2.8 | 265.2 | 20.2 | 21.2 | -8.9 | 122.7 |
| | **Outflow (m³)** | 24.1 | 22.6 | 0.0 | 190.2 | 33.2 | 23.9 | 0.0 | 92.8 | 20.5 | 20.2 | 0.0 | 90.4 |
| | **Volume (%)** | 60.6 | 19.4 | 21.7 | 93.4 | 61.0 | 22.2 | 24.2 | 95.5 | 55.7 | 24.0 | 22.5 | 96.8 |
| **Belesar** | **Inflow (m³)** | 83.3 | 125.5 | -10.4 | 1646.6 | 87.9 | 131.6 | -17.9 | 973.2 | 59.8 | 86.9 | -12.4 | 578.9 |
| | **Outflow (m³)** | 83.3 | 104.1 | 0.0 | 1819.2 | 86.9 | 101.6 | 0.0 | 920.1 | 58.8 | 65.2 | 4.2 | 578.3 |
| | **Volume (%)** | 57.1 | 26.0 | 4.6 | 96.6 | 64.6 | 23.0 | 20.7 | 93.9 | 54.7 | 22.3 | 21.4 | 94.4 |
| **Castrelo** | **Inflow (m³)** | 254.6 | 290.1 | -1.9 | 4054.5 | 317.5 | 295.5 | 28.6 | 2714.1 | 183.3 | 152.3 | 25.6 | 967.9 |
| | **Outflow (m³)** | 254.6 | 292.2 | 0.0 | 4243.5 | 317.6 | 299.2 | 0.0 | 2678.5 | 183.3 | 154.2 | 0.0 | 973.1 |
| | **Volume (%)** | 85.3 | 4.8 | 48.9 | 98.7 | 83.4 | 5.3 | 64.7 | 96.7 | 85.1 | 4.2 | 67.4 | 95.8 |
| **Frieira** | **Inflow (m³)** | 281.7 | 335.6 | 3.6 | 4621.4 | 347.3 | 353.5 | 8.0 | 2894.2 | 197.6 | 184.9 | 8.1 | 1163.9 |
| | **Outflow (m³)** | 281.7 | 335.9 | 29.2 | 4603.7 | 347.3 | 354.3 | 33.7 | 2927.2 | 197.5 | 185.3 | 31.7 | 1170.9 |
| | **Volume (%)** | 90.7 | 3.1 | 62.1 | 98.1 | 88.7 | 3.5 | 75.7 | 97.9 | 87.0 | 3.3 | 77.6 | 96.2 |
| **Peares** | **Inflow (m³)** | 88.6 | 117.1 | -0.4 | 1908.1 | 94.5 | 109.0 | -8.2 | 957.8 | 63.7 | 71.1 | 3.6 | 613.6 |
| | **Outflow (m³)** | 88.6 | 117.2 | 0.0 | 1918.9 | 94.5 | 108.8 | 0.0 | 962.7 | 63.7 | 71.0 | 6.3 | 622.7 |
| | **Volume (%)** | 93.0 | 12.4 | 26.1 | 99.5 | 97.4 | 1.3 | 88.2 | 98.8 | 95.1 | 5.0 | 73.6 | 98.9 |
| **San Martinho** | **Inflow (m³)** | 81.8 | 103.2 | 0.3 | 1213.4 | 112.9 | 129.0 | 8.0 | 1234.5 | 60.6 | 55.5 | 6.8 | 457.3 |
| | **Outflow (m³)** | 81.8 | 103.2 | 3.0 | 1213.2 | 112.9 | 129.1 | 8.3 | 1238.7 | 60.6 | 55.4 | 7.7 | 457.7 |
| | **Volume (%)** | 97.2 | 4.6 | 54.5 | 105.1 | 95.8 | 3.3 | 71.3 | 100.0 | 95.0 | 3.7 | 71.3 | 99.9 |
| **Santo Estevo** | **Inflow (m³)** | 145.4 | 185.3 | 1.6 | 1949.5 | 192.3 | 199.5 | 6.9 | 2000.5 | 103.0 | 86.4 | 14.2 | 724.7 |
| | **Outflow (m³)** | 145.6 | 186.3 | 0.0 | 1953.1 | 192.2 | 202.0 | 0.4 | 1999.6 | 103.0 | 87.9 | 11.6 | 573.0 |
| | **Volume (%)** | 85.7 | 13.1 | 30.4 | 99.9 | 84.2 | 8.7 | 57.8 | 98.0 | 82.2 | 9.7 | 58.1 | 96.7 |
| **Velle** | **Inflow (m³)** | 249.9 | 289.3 | -0.2 | 4624.7 | 311.0 | 281.4 | 18.5 | 2713.5 | 177.3 | 142.7 | 19.5 | 906.8 |
| | **Outflow (m³)** | 249.9 | 289.5 | 0.0 | 4637.2 | 311.0 | 281.7 | 27.0 | 2714.1 | 177.3 | 143.1 | 25.1 | 916.5 |



| | Volume (%) | 81.3 | 4.0 | 60.4 | 96.0 | 79.1 | 4.9 | 65.0 | 92.4 | 77.5 | 4.6 | 57.7 | 90.1 |
|---|---|---|---|---|---|---|---|---|---|---|---|---|---|

**Table 3. Statistical performance of the different models developed to predict the outflow.**

| Model | Reservoir | Train | | | | Validation | | | | Test | | | |
|---|---|---|---|---|---|---|---|---|---|---|---|---|---|
| | | r | NSE | RSR | Pbias | r | NSE | RSR | Pbias | r | NSE | RSR | Pbias |
| **Baseline** | Barcena | 0.933 | 0.866 | 0.366 | 0.019 | 0.921 | 0.844 | 0.394 | 0.025 | 0.925 | 0.852 | 0.385 | 0.044 |
| | Belesar | 0.921 | 0.842 | 0.398 | 0.015 | 0.928 | 0.857 | 0.378 | -0.030 | 0.908 | 0.817 | 0.428 | -0.022 |
| | Castrelo | 0.930 | 0.861 | 0.372 | 0.006 | 0.924 | 0.849 | 0.388 | 0.012 | 0.924 | 0.849 | 0.388 | -0.016 |
| | Frieira | 0.932 | 0.865 | 0.367 | 0.003 | 0.930 | 0.862 | 0.371 | 0.011 | 0.938 | 0.879 | 0.348 | 0.000 |
| | Peares | 0.934 | 0.869 | 0.362 | 0.011 | 0.939 | 0.880 | 0.347 | -0.017 | 0.922 | 0.845 | 0.394 | -0.033 |
| | San Martinho | 0.928 | 0.857 | 0.379 | 0.008 | 0.933 | 0.868 | 0.363 | 0.000 | 0.943 | 0.887 | 0.336 | 0.017 |
| | Santo Estevo | 0.941 | 0.883 | 0.342 | -0.004 | 0.925 | 0.852 | 0.385 | 0.039 | 0.916 | 0.834 | 0.407 | -0.049 |
| | Velle | 0.931 | 0.862 | 0.371 | 0.006 | 0.939 | 0.879 | 0.347 | 0.017 | 0.936 | 0.874 | 0.356 | -0.037 |
| | **Average** | **0.931** | **0.863** | **0.370** | **0.006** | **0.930** | **0.861** | **0.372** | **0.003** | **0.925** | **0.852** | **0.384** | **-0.017** |
| **Linear reg.** | Barcena | 0.938 | 0.880 | 0.346 | 0.000 | 0.926 | 0.859 | 0.376 | 1.673 | 0.919 | 0.844 | 0.395 | 2.730 |
| | Belesar | 0.940 | 0.884 | 0.341 | -0.053 | 0.950 | 0.903 | 0.311 | -1.007 | 0.926 | 0.857 | 0.378 | -4.328 |
| | Castrelo | 0.944 | 0.891 | 0.329 | -0.018 | 0.942 | 0.887 | 0.335 | 3.822 | 0.944 | 0.892 | 0.328 | -1.362 |
| | Frieira | 0.938 | 0.881 | 0.345 | -0.009 | 0.934 | 0.873 | 0.356 | 1.962 | 0.939 | 0.882 | 0.344 | -2.200 |
| | Peares | 0.932 | 0.869 | 0.361 | -0.085 | 0.940 | 0.885 | 0.339 | -0.148 | 0.923 | 0.853 | 0.383 | -3.545 |
| | San Martinho | 0.895 | 0.781 | 0.468 | -11.19 | 0.935 | 0.876 | 0.352 | 1.866 | 0.934 | 0.870 | 0.361 | -4.813 |
| | Santo Estevo | 0.941 | 0.885 | 0.339 | -2.584 | 0.935 | 0.876 | 0.353 | 2.253 | 0.932 | 0.869 | 0.361 | 0.569 |
| | Velle | 0.937 | 0.878 | 0.350 | 0.000 | 0.942 | 0.889 | 0.334 | 1.568 | 0.941 | 0.886 | 0.338 | -1.807 |
| | **Average** | **0.932** | **0.867** | **0.362** | **-1.992** | **0.940** | **0.884** | **0.340** | **1.474** | **0.934** | **0.873** | **0.356** | **-2.498** |
| **MLP** | Barcena | 0.941 | 0.884 | 0.341 | -2.520 | 0.931 | 0.867 | 0.365 | 0.023 | 0.923 | 0.851 | 0.386 | 0.012 |
| | Belesar | 0.948 | 0.899 | 0.318 | 0.337 | 0.955 | 0.911 | 0.298 | -0.685 | 0.930 | 0.863 | 0.370 | -3.206 |
| | Castrelo | 0.947 | 0.895 | 0.324 | -3.566 | 0.949 | 0.899 | 0.318 | -0.099 | 0.946 | 0.887 | 0.336 | -7.250 |
| | Frieira | 0.950 | 0.902 | 0.314 | 3.137 | 0.942 | 0.886 | 0.338 | 4.028 | 0.944 | 0.889 | 0.334 | 4.847 |
| | Peares | 0.947 | 0.896 | 0.322 | -3.399 | 0.943 | 0.887 | 0.337 | -4.896 | 0.928 | 0.855 | 0.381 | -6.530 |
| | San Martinho | 0.941 | 0.886 | 0.338 | 0.586 | 0.939 | 0.881 | 0.345 | 0.774 | 0.947 | 0.897 | 0.321 | -1.267 |
| | Santo Estevo | 0.950 | 0.902 | 0.313 | -1.930 | 0.941 | 0.886 | 0.337 | -0.744 | 0.937 | 0.875 | 0.354 | -2.314 |
| | Velle | 0.937 | 0.878 | 0.350 | 2.378 | 0.946 | 0.893 | 0.327 | 3.347 | 0.944 | 0.889 | 0.333 | 3.257 |
| | **Average** | **0.945** | **0.893** | **0.327** | **-0.622** | **0.943** | **0.889** | **0.333** | **0.219** | **0.937** | **0.876** | **0.352** | **-1.556** |
| **NARX** | Barcena | 0.947 | 0.897 | 0.320 | -0.006 | 0.937 | 0.877 | 0.351 | 0.642 | 0.925 | 0.855 | 0.381 | -0.624 |
| | Belesar | 0.954 | 0.911 | 0.298 | -0.214 | 0.953 | 0.908 | 0.303 | -0.201 | 0.934 | 0.870 | 0.360 | -2.687 |
| | Castrelo | 0.954 | 0.910 | 0.301 | -2.019 | 0.953 | 0.908 | 0.303 | -0.379 | 0.949 | 0.899 | 0.318 | -3.216 |




| | | | | | | | | | | | | |
|---|---|---|---|---|---|---|---|---|---|---|---|---|
| | **Frieira** | 0.946 | 0.894 | 0.325 | -1.803 | 0.945 | 0.892 | 0.329 | -0.413 | 0.945 | 0.894 | 0.326 | 0.572 |
| | **Peares** | 0.951 | 0.904 | 0.309 | 0.410 | 0.946 | 0.893 | 0.326 | -1.743 | 0.932 | 0.868 | 0.363 | -0.824 |
| | **San Martinho** | 0.946 | 0.896 | 0.323 | -0.761 | 0.942 | 0.886 | 0.337 | 0.723 | 0.953 | 0.908 | 0.304 | -3.209 |
| | **Santo Estevo** | 0.957 | 0.916 | 0.290 | -0.348 | 0.942 | 0.888 | 0.335 | 0.522 | 0.941 | 0.885 | 0.340 | 0.252 |
| | **Velle** | 0.944 | 0.891 | 0.330 | -0.083 | 0.951 | 0.904 | 0.311 | -0.224 | 0.945 | 0.892 | 0.328 | -2.880 |
| | **Average** | **0.950** | **0.902** | **0.312** | **-0.603** | **0.946** | **0.894** | **0.324** | **-0.134** | **0.941** | **0.884** | **0.340** | **-1.577** |
| | **Barcena** | 0.952 | 0.905 | 0.308 | 1.083 | 0.941 | 0.886 | 0.338 | 1.275 | 0.927 | 0.858 | 0.376 | 1.446 |
| | **Belesar** | 0.943 | 0.889 | 0.334 | 0.008 | 0.940 | 0.882 | 0.344 | -3.292 | 0.931 | 0.865 | 0.368 | -3.621 |
| | **Castrelo** | 0.949 | 0.899 | 0.318 | 1.222 | 0.950 | 0.901 | 0.315 | 2.717 | 0.947 | 0.896 | 0.323 | -0.581 |
| | **Frieira** | 0.946 | 0.896 | 0.323 | 0.286 | 0.946 | 0.894 | 0.325 | -0.190 | 0.945 | 0.893 | 0.326 | 1.183 |
| **LSTM** | **Peares** | 0.937 | 0.876 | 0.353 | 0.846 | 0.943 | 0.888 | 0.334 | -2.935 | 0.928 | 0.857 | 0.378 | -1.683 |
| | **San Martinho** | 0.947 | 0.897 | 0.321 | 0.868 | 0.932 | 0.869 | 0.362 | 0.153 | 0.950 | 0.898 | 0.320 | -2.251 |
| | **Santo Estevo** | 0.955 | 0.912 | 0.297 | 0.692 | 0.945 | 0.892 | 0.328 | -0.107 | 0.933 | 0.866 | 0.367 | 0.979 |
| | **Velle** | 0.940 | 0.882 | 0.343 | 1.239 | 0.947 | 0.897 | 0.322 | 0.006 | 0.944 | 0.889 | 0.333 | -1.992 |
| | **Average** | **0.946** | **0.894** | **0.325** | **0.781** | **0.943** | **0.889** | **0.334** | **-0.297** | **0.938** | **0.878** | **0.349** | **-0.815** |