# Peer review of "Comparison of machine learning techniques for reservoir outflow forecasting"

_Natural Hazards and Earth System Sciences, 2022_

## Referee Comment (RC3)

** | **0.946** | **0.894** | **0.325** | **0.781** | **0.943** | **0.889** | **0.334** | **-0.297** | **0.938** | **0.878** | **0.349** | **-0.815** |

[referee-annotated manuscript omitted]

---

## Author Comment (AC3)

First of all, we would like to thank the reviewer for the valuable comments that undoubtedly contributed to improve this work and make it more useful for the community.

Hereafter the questions and suggestions raised by the reviewer will be addressed:

> *The paper is well written and the figures are of high quality.*
>
> *The research is well structured and explained: the problem, the methods, the data and the results.*
>
> *However, the interest of the topic is not clear. The input variables considered exclude essential information on the reservoir outflow, which limits the predictive accuracy for any algorithm. Operation strategies play a key role in the outflow values. During dry periods, only the environmental flow is probably discharged in dams without hydropower units. In other situations, the flow will be obviously influenced by the strategy applied for optimizing power generation. Other restrictions such as freeboards in wet periods also have an influence.*

Certainly, the operation strategies of the dam are key to determine its outflow. They can be used in different ways like developing a rule-based system or incorporating them into a hybrid machine learning model. However, this research has focused on the capabilities of different machine learning approaches to forecast the total outflow of the next 24 hour and therefore infer the operation strategies from data.

> *Predicting reservoir outflow without considering this information is difficult, and the use of one technique or the other has a lower effect in accuracy.*
>
> *The authors actually mention this in the introduction:*
>
> *"it can be a good approximation in flood scenarios during wet seasons, especially in small reservoirs or when they are nearly full and have little margin to alter the natural flow of the river."*
>
> *The results confirm this intuition:*
>
> *"Looking more closely at the data in Table 3 a tendency is detected in reservoirs with a higher capacity to have worse statistics than those of lower capacity."*
>
> *I suggest modifying the statement in the introduction. I would rather say that the proposed approach can only be applied with reasonable accuracy to small reservoirs during wet seasons in wet climatic conditions. All reservoirs considered are located in Galicia, an area with higher rainfall rates than other Spanish regions. The approach is probably less applicable in dry regions. This is an important piece of information for the community.*

Following the reviewer indications, the introduction was modified accordingly to clarify to the reader that the results shown in this research may not be extrapolated to areas with different characteristics:

*…However, it is worth noting that the results obtained in this study may not be completely extrapolated to other areas with larger reservoirs and/or dry climatic conditions.*

> ***The authors may consider performing some kind of variable selection. For instance, adding inputs such as the season, the month or the day of the year, could serve as proxies to the operation rules. Also, the gradient of pool level could be informative (the operation of the reservoirs is probably different for increasing than for decreasing pool level for a given value of the stored volume). This may result to be more useful for increasing accuracy than the use of some specific, complex ML algorithm: the results show that differences among algorithms are mostly negligible and that even a very simple regression (MLR) is comparable to sophisticated techniques.***

At a preliminary stage, some temporal variables like the month or the day were considered but not significant improvement was obtained. This may be caused by the fact that the transition from dry to wet seasons varies from year to year.

Regarding to adding the gradient of the pool level as an input variable, it may improve the results of MLR and MLP techniques as it adds extra information, however recurrent neural networks may not get significant benefit from this extra variable as they already incorporate data from previous timesteps.

We have updated the manuscript accordingly as this information that may be useful to the reader:

*Although the validity of these models was assessed using only a limited range of input variables, there are many other variables that influence the functioning of the reservoirs, such as the weather forecast or the electric power demand. Also, it is possible that the addition of certain input variables like the gradient of the pool level could close the gap of MLR or MLP methodologies with recurrent neural networks. It is being studied how to incorporate these variables into these models and if they can add any improvement to the predictions.*

> ***Nonetheless, operation strategy is the essential element in outflow. Overall, the usefulness of predicting outflows from reservoirs without information on the operation strategy is questionable. Every river basin authority should have such information available. The authors should clearly explain the scope of application of the approach, i. e., under which conditions the operation strategy is not known by the water resources management authority.***

There are several scenarios where the operation policies may not be available. Sometimes the river basin is divided in different countries. Also, the operation of each dam often depends on private companies and their operating strategy is only available under a non-disclosure agreement and the results based on this information may not be publicly published. Even if the operating strategy is available, some of the variables may not be available at the moment of making the prediction. This clarification is made in the manuscript:

*Is not unusual for rivers to pass through different countries or administrative regions with different policies and regulations. It is also common for dams to be operated by private companies with different operating policies and interests. The operation of these structures depends not only on natural factors and well-defined operating rules but also on external demand. These aspects can also hinder the access or the utilization of the operation rules of the reservoir. This adds a significant amount of uncertainty in predicting the outflow of a reservoir at any given time, making it difficult to incorporate into physics-based models, which is a disadvantage in water resource management and flood risk prevention.*

**The pdf file attached includes additional comments and suggestions.**

The responses to the comments included in the pdf are detailed below:

**"Therefore, the lamination capacity of the dam is almost cancelled, and the outflow is equal to the inflow."**

**This is not correct: even at 100% storage (pool level equals the spillway level), the reservoir capacity above such level is far from negligible. On the contrary, the reservoir volume is higher for high pool levels. This is only true in small reservoirs with high inflow rates, after long period of high inflows, close to the spillway capacity.**

**"Although this approach is an over-simplification of river dynamics, it can be a good approximation in flood scenarios during wet seasons, especially in small reservoirs or when they are nearly full and have little margin to alter the natural flow of the river."**

**I would say that only in these situations (small reservoirs, wet seasons), the former condition can be considered valid**

The paragraph was updated accordingly to the comments:

*In order to forecast the outflow of a reservoir, the most simplistic approach involves assuming that the reservoir surpasses its storage capacity and therefore the outflow is equal to the inflow. Although this approach is an over-simplification of river dynamics, it can be a reasonable approximation only in relatively small reservoirs during wet season when they are close to the spillway capacity after a period of high inflows, and therefore have little margin to alter the natural flow of the river.*

**"For all the models, the inflow, outflow and volume percent values are used as input data to predict the outflow of the next day."**

**Some exploratory plots would be useful (e.g., correlation plots)**

A new figure was added with heatmaps of the correlation between variables for each of the reservoir:

[Figure]

*Figure 1. Correlation heatmaps for each of the analyzed reservoirs.*

*Figure 2 shows the correlation between the variables involved for each of the analyzed reservoirs. It can be observed that there is a strong correlation between the inflow and the outflow variables. However, the two largest reservoirs analyzed (Barcena and Belesar) show a lower correlation between inflow and outflow, that it is in concordance with their higher capacity and lower average occupation. The correlation between volume percent and the outflow is low all the cases but slightly higher on Barcena and Belesar.*

**"In this sense, the number of neurons in the input layer is fixated by the number of variables (inflow, outflow and volume (%) at day d) that will be used to try to predict the desired variable (outflow for day d+1)."**

**Please, provide some reference to support this statement. According to Hastie et al [1], "Typically the number of hidden units is somewhere in the range of 5 to 100, with the number increasing with the number of inputs and number of training cases." This means that there should be some relation between number of inputs and number of neurons, not that both quantities should be equal.**

**[1] Hastie, T., Tibshirani, R., Friedman, J. H., & Friedman, J. H. (2009). The elements of statistical learning: data mining, inference, and prediction (Vol. 2, pp. 1-758). New York: springer.**

The sentence refers to the input layer of the ANN that depends on the number of input variables (see Figure 2 in the original manuscript). We have rewritten the paragraph to clarify this:

*As previously said, the first ANN models developed were feed-forward neural networks with a back propagation algorithm. In this kind of ANNs, the information passes through different layers. In the input layer, the information is received from the database, and it is sent to the hidden layer where the information is treated. Finally, this new information is sent to the output layer where a result is generated. The number of neurons in the input layer is determined by the number of input variables (inflow, outflow and volume (%) at day d) that will be used to try to predict the desired variable (outflow for day d+1). In this methodology, only one hidden layer was used, and the number of neurons was determined by the trial-and-error method (being studied between one and seven). Finally, in the output layer, there will be as many neurons as variables to be predicted (in this case, one). The number of cycles was studied between 1 and 131,072 in 17 steps with a logarithmic or lineal scale, and the decay parameter was used to decrease the learning rate during the learning process (true or false). The best MLP model developed (lineal or logarithmic scale) was selected based on the lowest RMSE value for the validation subset.*

> **"In the output layer, there will be as many neurons as variables to be predicted (in this case, one). Finally, in this research, only one hidden layer was used, and the number of neurons was determined by the trial-and-error method (being studied between one and seven)."**
>
> **This sounds incoherent with the previous statement. Please, clarify.**

The first sentence mentions the output layer of the ANN that depends on the number of output variables. The second sentences refer to the hidden layer whose number of neurons has been determined by trial-and-error ranging values from 1 and 7. We have rewritten the paragraph to clarify this:

*As previously said, the first ANN models developed were feed-forward neural networks with a back propagation algorithm. In this kind of ANNs, the information passes through different layers. In the input layer, the information is received from the database, and it is sent to the hidden layer where the information is treated. Finally, this new information is sent to the output layer where a result is generated. The number of neurons in the input layer is determined by the number of input variables (inflow, outflow and volume (%) at day d) that will be used to try to predict the desired variable (outflow for day d+1). In this study, only one hidden layer was used, and the number of neurons was determined by the trial-and-error method (being studied between one and seven). Finally, in the*

*output layer, there will be as many neurons as variables to be predicted (in this case, one). The number of cycles was studied between 1 and 131,072 in 17 steps with a logarithmic or lineal scale, and the decay parameter was used to decrease the learning rate during the learning process (true or false). The best MLP model developed (lineal or logarithmic scale) was selected based on the lowest RMSE value for the validation subset.*

The figure of the architecture of a MLP was also modified using a variable nomenclature homogeneous with the rest of the paper:

[Figure]

***Figure 2. Architecture of a multilayer perceptron with three inputs, a hidden layer with four neurons and a single output. The input variables $y_d$, $r_d$ and $v_d$ are the measured outflow, inflow, and volume percent for day d, meanwhile the output variable $\hat{y}_{d+1}$ is the forecasted outflow for day d+1.***

**From my point of view, this result may not be general, i.e., NARX may underestimate peak flows in other settings and vice-versa**

The following sentence was added to the results:

*On the opposite, in the lower flows on the dry season, the NARX model was the best performer providing accurate predictions, however, the LSTM model had some difficulties at very low flow rates, especially in the summers of 2017 and 2019. This makes the LSTM model less suitable for water management systems where the accuracy on the dry season is essential for a better exploitation of the water resources. In any case, these results may not be generalized and they can differ in other scenarios.*

**Do the authors consider MLR as an ML technique? Please, explain and justify.**

Linear regression and multivariate linear regression are widely known machine learning techniques based on supervised learning and that fits the definition provided in the introduction. In a MLR model the coefficients are adjusted to find the best fit to the training data. This adjustment or learning can be performed by many algorithms like the gradient-descent or the least-squares methods. There are many works on machine learning and linear regression available in the literature, for example:

Potok, T. (2021). Adiabatic quantum linear regression. Scientific reports, 11(1), 1-10.

Behera, S., & Prathuri, J. R. (2020, November). Application of homomorphic encryption in machine learning. In 2020 2nd PhD Colloquium on Ethically Driven Innovation and Technology for Society (PhD EDITS) (pp. 1-2). IEEE.

***The size of the dataset may have an influence, but the linear/nonlinear nature of the phenomenon is probably the main reason of the difference between MLR and ANN***

The following sentence was added to clarify this aspect:

*ANN is a more suitable method than MLR, given the non-linear nature of the phenomenon. Since the ANN-based models were able to outperform the more traditional MLR models, it can be concluded that the number of samples in the dataset are within the limit for training an ANN, albeit larger datasets would possibly lead to better models especially when dealing with extreme events.*

***Please, comment on the limitations of the approach for dry areas/seasons.***

A comment on the limitations was added to the conclusions:

*The overall observations confirm that the analysed ML models are capable to predict the outflow of reservoirs and, therefore, can be incorporated into different systems such as water resource management systems or early warning systems. However, the results obtained in this research are limited to relatively small reservoirs located in wet areas and may not be extrapolated to larger reservoirs or dry areas, requiring additional research.*

---

## Author Response (AR2)

We would like to thank the editor for the valuable comments. Hereafter we hope to clarify the questions arised:

**1) Regarding the relevance of the analyses w.r.t. the influence of the operation strategies on the outflow values: please reflect on the importance of your work compared to such operational conditions in the manuscript (now it is only given in the response to the reviewer).**

Now we have included this clarification in the text (lines 88-91):

*However, it is worth noting that the results obtained in this study may not be completely extrapolated to other areas with larger reservoirs and/or dry climatic conditions. Also, it is important to clarify that the operation strategies of the dam are key to determine its outflow and they can be used in different ways to improve the prediction models. However, this research focuses on the capabilities of different machine learning approaches to forecast the total outflow of the next 24-hour period and therefore infer the operation strategies from the data.*

**2) The question regarding your assumption that the lamination capacity of the dam is almost cancelled. I.e.**
**"Therefore, the lamination capacity of the dam is almost cancelled, and the outflow is equal to the inflow."**
**This is not correct: even at 100% storage (pool level equals the spillway level), the reservoir capacity above such level is far from negligible. On the contrary, the reservoir volume is higher for high pool levels. This is only true in small reservoirs with high inflow rates, after long period of high inflows, close to the spillway capacity.**

We wanted to apologize for not clarifying this point sufficiently in the previous revision. We agree with Referee #3, even at 100% storage the reservoirs retain certain capacity above that level. Therefore, we removed that statement in the previous revision (lines 61-63):

*In order to forecast the outflow of a reservoir, the most simplistic approach involves assuming that the reservoir  surpasses its storage capacity., and therefore the outflow is equal to the inflow.*

And clarified that it can be acceptable only under certain conditions as the referee remarked (lines 64-66):

*…it can be a  reasonable approximation only in  relatively small reservoirs  during wet season when they are  close to the spillway capacity after a period of high inflows, and therefore have little margin to alter the natural flow of the river.*

However, we wanted to remark that this is only a possible approach that is not even studied in this work, and simply consists in considering that the river flows at a natural regime as

there were no dams. We have modified the paragraph in this iteration in order to clarify this aspect (lines 62-68):

*In order to forecast the outflow of a reservoir, the most simplistic approach involves assuming that  the outflow  would be equal to the inflow. Although this approach is an over-simplification of river dynamics, it can be a reasonable approximation under very specific conditions. For example, in relatively small reservoirs during wet season when they are close to the spillway capacity after a period of high inflows, and therefore have little margin to alter the natural flow of the river.  Another simplistic approximation  would be to  assuming that the outflow of the reservoir for a given day d will be the same as on day d-1.*